# Is project-based learning effective among kindergarten and elementary students? A systematic review

**Marta Ferrero**[1], **Miguel A. Vadillo**[2]*, **Samuel P. León**[3]*

**1** Departamento de Investigación y Psicología de la Educación, Universidad Complutense de Madrid, Madrid, Spain, **2** Departamento de Psicología Básica, Universidad Autónoma de Madrid, Madrid, Spain, **3** Departamento de Pedagogía, Universidad de Jaén, Jaén, Spain

* miguel.vadillo@uam.es (MAV); sparra@ujaen.es (SPL)

## Abstract

Project-based learning (PjBL) is becoming widespread in many schools. However, the evidence of its effectiveness in the classroom is still limited, especially in basic education. The aim of the present study was to perform a systematic review of the empirical evidence assessing the impact of PjBL on academic achievement of kindergarten and elementary students. We also examined the quality of studies, their compliance with basic prerequisites for a successful result, and their fidelity towards the key elements of PBL intervention. For this objective, we conducted a literature search in January 2020. The inclusion criteria for the review required that studies followed a pre-post design with control group and measured quantitatively the impact of PBL on content knowledge of students. The final sample included eleven articles comprising data from 722 students. The studies yielded inconclusive results, had important methodological flaws, and reported insufficient or no information about important aspects of the materials, procedure and key requirements from students and instructors to guarantee the success of PjBL. Educational implications of these results are discussed.

## Introduction

Over the last decade, numerous institutions have addressed the skills and dispositions that are expected to be vital for schooling in 21st century. Some of these skills are critical thinking, communication, collaboration, or creativity [1,2]. According to many experts, although the prevailing methods of direct instruction and recitation may be effective for the acquisition of factual knowledge, these skills demand new pedagogical approaches [3]. Within this context, *project-based learning* (PjBL) and *problem-based learning* (PBL) have emerged as valuable inquiry approaches to achieve the so-called skills for the 21st century [4].

PjBL and PBL are usually described as active, student-centred methods of instruction that encourage students to work in collaborative groups on real-world questions or challenges to promote the acquisition of higher-order thinking skills, while teachers act as facilitators of learning [4–14]. Despite these common characteristics, PjBL and PBL also present some

**Data Availability Statement:** All relevant data are reported within the paper. As the present systematic review does not include a quantitative meta-analysis, there are no additional datafiles,

beyond the information reported in the results section.

**Funding:** Funded by Agencia Estatal de Investigación with grant number PSI2017-85159-P (MAV), Comunidad de Madrid (ES) with grant number 2016-T1/SOC-1395 (MAV) and Comunidad de Madrid (ES) with grant number 2020-5A/SOC-19723 (MAV). Funders did not play any role in any phase of the study.

**Competing interests:** The authors have declared that no competing interests exist.

noticeable differences. For instance, in PjBL learners are expected to follow correct procedures towards a desired end-product or presentation during which they are likely to encounter different problems [7,9,14], while in PBL the emphasis is on the role of the students to define the problem and develop a solution [9,15,16]. In addition, while in PBL the solution to the problem is merely suggested, in PjBL it must be executed [7]. Finally, PjBL occurs over an extended time period, while PBL normally lasts a few days [5]. In practice, given the usual difficulties in distinguishing one from the other or in defining their key features [5,9,14], both terms are often employed interchangeably among researchers [4,14] and teachers [17]. Since both approaches are closely related and share a central end, throughout this review we will use the term PjBL to refer to both of them.

PjBL originated in an architecture school in Rome in the 16th century [18]. Forced by organizational and curricular constraints, lectures were moved to weekends and, to minimize the potential lack of motivation among students, teachers decided to use this approach. Later on, dissatisfaction with standard methods in medical education led a large number of medical schools to adopt PjBL [6], which progressively extended to different undergraduate studies [10,15,19]. The main reasons for adopting this approach were student disenchantment and boredom caused by the vast amount of information they had to learn with presumably little impact on daily practice [6]. In general, the quantitative reviews performed in medical schools show that the traditional approach to learning in the classroom outperforms PjBL in the acquisition of basic science knowledge, while, conversely, PjBL is superior to the traditional approach when it comes to learning clinical problem solving, that is, application of knowledge [8,20–23] and ability to link concepts [19,24]. More generally, different studies conducted with undergraduate students have shown that PjBL can help students improve academic achievement [25] and build flexible knowledge [10].

In spite of the promising results of PjBL, some authors have drawn attention to the existing gaps in our knowledge about the conditions under which PjBL can be more beneficial than other approaches [21]. Similarly, researchers have outlined the importance of considering some prerequisites necessary for students and teachers to be successful in higher education when using PjBL. In the case of students, these requisites include the previous acquisition of basic content knowledge about the target problem or project and competence in some learning strategies and skills (i.e., the ability to communicate ideas effectively). For teachers, the requisites include, for instance, proficiency in appropriate teaching strategies and tools (i.e., the provision of adequate scaffolding). If these prerequisites are not met, students might not benefit from PjBL and teachers might not be able to apply it with any guarantee of success [7]. Finally, due to the various ways in which PjBL has been implemented in the classroom, it is important to pay attention to the fidelity with which its main principles are applied when evaluating its impact on learning. Ideally, an intervention faithful to the PjBL approach should include all its essential components as defined in the literature. Otherwise, there is a risk of attributing the (positive or negative) effects of an intervention to PjBL when, in fact, the intervention does not meet the definition of PjBL. As mentioned above, some of the central elements to PjBL are the need of a problem to drive the activities and a final artifact or product; the use of group work methodology; the empowerment of students; the provision of guidance and resources by teachers; and the adoption of evaluation tools adapted to PjBL characteristics (i.e., notebook entries or portfolio).

The effectiveness of PjBL has also been tested at the secondary school level, although to a lesser extent than in medical schools and undergraduate studies. As in the case of undergraduate settings, this approach has been shown to improve the academic achievement of secondary school students in different subjects, such as economics [26,27], history [28], or STEM (Science, Technology, Engineering, and Mathematics) [29–31]; for a review, [5,25,32]. In spite of these

promising results, some researchers have warned of the limited number of scientific studies on PjBL instruction in high school and emphasize the need for more and better research before strong claims can be made about the potential benefits of this approach [5,25,33]. Furthermore, most of the studies conducted to date followed quasi-experimental designs, so the existing evidence on the impact of PjBL in secondary school level appears to be weak [34].

At present, a growing number of kindergarten and primary schools are introducing PjBL in their classrooms. Even more, in countries like Spain, the educational authorities of some regions have made the inclusion of PjBL in classroom programmes mandatory [35]. Considering the good results obtained in higher levels, it is reasonable to expect that PjBL would also contribute to promoting the learning of kindergarten and primary students. Nevertheless, due to the considerable differences between senior and novice learners [36], this assumption deserves further analysis. Unlike the cases mentioned above, there is still no systematic review on the efficacy of PjBL exclusively focused on these basic levels of education. To our knowledge, there are two non-systematic reviews and one meta-analysis that have addressed the effectiveness of PjBL in different levels, including to some extent kindergarten and primary education. The first one focuses on the effect of PjBL in students from kindergarten to K-12. It includes both quantitative and qualitative studies [11]. The second one is an overview of the effectiveness of PjBL from preschool to higher education and pre-service teacher training [12]. And the third one analyses quantitatively the impact of PjBL on academic achievement in comparison with traditional teaching from third grade elementary school to senior college students and explores what study features might moderate this effect [25]. Overall, these studies conclude that PjBL is an effective means of teaching content information. However, in all cases, important pieces of information are missing from the studies analysed. For instance, none of the reviews assess the level of student and instructor compliance with the basic requirements of PjBL. Similarly, the fidelity of interventions to the main principles of PjBL is not analysed. Finally, only one of the studies [11] analyses the information related to the quality of the primary studies. Considering that the authors of these reviews have highlighted the need of better and more detailed research, it seems advisable to report and discuss this type of information more thoroughly. Without this information, it is difficult, if not impossible, to draw firm conclusions about the effectiveness of PjBL for kindergarten and elementary school students.

The main objective of the present study was to perform a systematic review on the effect of PjBL on the acquisition of content knowledge in kindergarten and primary students, including as much relevant information as possible on methodological and conceptual aspects. Specifically, we examined the quality of existing studies, their compliance with basic prerequisites for a successful PjBL intervention, and the fidelity to the interventions in light of the key elements of PjBL, as reported in the literature.

## Method

### Search procedures

The present systematic review follows the PRISMA recommendations. On January 23th 2020 the first author (MF) performed an electronic search on the *Web of Science*, *PsycInfo*, and *ERIC* entering the terms "(project based OR problem based) AND (learning OR intervention OR approach OR instruction)" into the Topic field. The search was limited to (a) articles in English, (b) published between 1900 and 2020, (c) with categories restricted to "education/educational research" and "psychology". Unpublished dissertations, reviews and meta-analyses were excluded at this stage. After removing 523 duplicates this initial search yielded a sample of 34,246 studies.

The titles and abstracts of these studies were screened by MF using the inclusion criteria c1-c5 explained below. This resulted in the exclusion of 32,208 studies that did not meet the inclusion criteria. MF and SPL independently read the full text of the remaining 38 studies to verify that they fulfilled criteria c1-c5. Among the initial set of 38 articles assessed for eligibility, nine articles met the inclusion criteria. Thereupon, we performed descendancy searches of articles citing or cited by these nine papers to identify additional studies. The titles and abstracts of the second search were screened by MF and this resulted in 16 full-text articles that were also independently read by MF and SPL. No additional study was selected from this set. Finally, on request of an anonymous reviewer, we added two extra studies included in a meta-analysis. Therefore, the final sample of articles reviewed for inclusion comprised eleven articles (see Table 1) [37–47]. Fig 1 shows a PRISMA flowchart summarizing the literature search process. Across all the full-text articles read for inclusion, the initial inter-rater agreement was 98.31%. Disagreements were resolved by discussion and consensus between the two researchers until there was 100% agreement.

## Selection criteria

The studies were only included if they met the following criteria: c1) the aim was to evaluate the effect of PjBL on content knowledge; c2) they followed a pre-post design with control group; c3) the target sample comprised students from kindergarten to grade 6; c4) they were written in English; and c5) they were peer-reviewed. Therefore, narrative and systematic reviews, doctoral dissertations, posters, registered study protocols, commentaries, books and book chapters, essays, and other theoretical reports were excluded from the review.

## Data extraction and coding

The eleven studies that met the inclusion criteria were independently examined in depth and coded by MF and SPL. They recorded information related to general aspects (authors, year of publication, and journal), participants (country of origin, sample size, age, educational level, and school type), method (design, duration, dependent variable, and measuring tools), and the main results obtained by each study.

In order to overcome important shortcomings of the reviews mentioned above, we used the quality scale developed by [48] with just one modification (see below). Very briefly, the original 17-item scale includes information related to the quality of various methodological aspects of an empirical research such as randomisation, blinding, replicability, or test validity (see Fig 2). Each item could be assigned three values: positive, negative, and unknown. For each study, MF and SPL independently assigned a value to each item, reaching an initial agreement of 98.30%. Disagreements were resolved through discussion until 100% consensus was reached. Fig 2 shows the values assigned to each item and study.

The quality scale used in [48] was originally created to assess educational interventions inspired by the multiple intelligences theory. Unlike research in that field, the PjBL literature offers a wealth of information on the basic prerequisites that both students and instructors should meet for PjBL to be successful, as well as on the key principles that characterize this approach. Therefore, for this study, we removed Item 6 from the original scale (referring to intervention fidelity) and replaced it by a full new scale intended to analyze both the compliance of teachers and students with the basic prerequisites of PjBL and the fidelity of the intervention to the principles underlying this method. This new scale consists of 30 items divided in two parts. Part A refers to the prerequisites and Part B refers to intervention fidelity. The 14 items in Part A are grouped into three categories. Items a1 to a6 belong to the category "Previous training of students in group work", Item a7 to "Measurement of prior knowledge of students", and Items a8 to a14 to "Teacher training in PjBL". The 12

**Table 1. Articles that met inclusion and quality criteria.**

| Authors, year | Country | Sample size (E/N) | Age (mean) | Sample type | Educational level | School type | Instructor | Duration | Dependent variable | Tests to measure DV | Results |
|---|---|---|---|---|---|---|---|---|---|---|---|
| Alacapinar, 2008 [37] | Turkey | 42 (21, 21) | (11.4 years) | normal population | 5th grade | n.s. | n.s. | n.s. | Cognitive domain | n.s. | PBL group outperformed significantly control group. |
| Aral et al., 2010 [38] | Turkey | 28 (14, 14) | 6 year | normal population | Preschool education | n.s. | n.s. | 12 weeks (1 day per week) | Children's conceptual development and school readiness composite | Bracken Basic Concept Scale-Revised | No differences between PBL and control group. |
| Aslan, 2013 [39] | Turkey | 47 (24, 23) | 6 year | normal population | Preschool education | public school | Teacher | 12 weeks (3 days per week) | Categorization skills | A categorization test | PBL group outperformed significantly control group. |
| Çakici et al., 2013 [46] | Turkey | 44 (22, 22) | n.s. | normal population | 5th grade | public school | Teacher and researcher | 5 weeks | Sciences knowledge | The Light and Sound Achievement Test | PBL group outperformed significantly control group. |
| Can et al., 2017 [40] | Turkey | 26 (17, 9) | 6 year | normal population | Preschool education | n.s. | Teacher | 32 weeks | Scientific process skills and conceptions | Preschool Scientific Process Skills Scale | No comparison reported between PBL and control group. |
| Gültekin, 2005 [41] | Turkey | 40 (20, 20) | n.s. | normal population | 5th grade | n.s. | n.s. | 3 weeks (6 hours per week) | Achievement in social studies | An achievement test | No quantitative data reported. |
| Hastie et al., 2017 [42] | EEUU | 185 (109, 76) | (10.6 years) | normal population | 5th grade | rural school | Teacher and researcher | 9 week | Fitness knowledge | Fitness Knowledge Test | PBL group outperformed significantly control group. |
| Karaçalli et al., 2014 [43] | Turkey | 143 (73, 70) | 9–11 years | normal population | 4th grade | n.s. | Teacher and researcher | 4 weeks | Sciences knowledge | Electricity in Our Life Achievement Test (ELACH), Science Course Attitude Scale (ELATT) | PBL group outperformed significantly control group. |
| Kucharski et al., 2005 [47] | EEUU | 61 (30, 31) | n.s. | normal population | 1st, 3th and 4th grade | n.s. | Teacher | n.s. | Sciences knowledge | Terra Nova Scale | PBL group outperformed significantly control group (except in 4th grade). |
| Lin, 2015 [44] | Taiwan | 56 (28, 28) | 11 years | normal population | 5th grade | public school | Teacher | 12 weeks (40 min per week) | Vocabulary knowledge | Vocabulary knowledge test | No differences between PBL and control group. |
| Zumbach et al., 2004 [45] | Germany | 50 (24, 26) | (10.1 years) | normal population | 4th grade | n.s. | Teacher and computer | n.s. | Forest animals knowledge | A konowledge test | No differences between PBL and control group in the short-term but yes in the long-term for PBL group. |

Note: DV: Dependent variable. (E) Experimental Group. (C) Control Group. n.s.: Not specified.

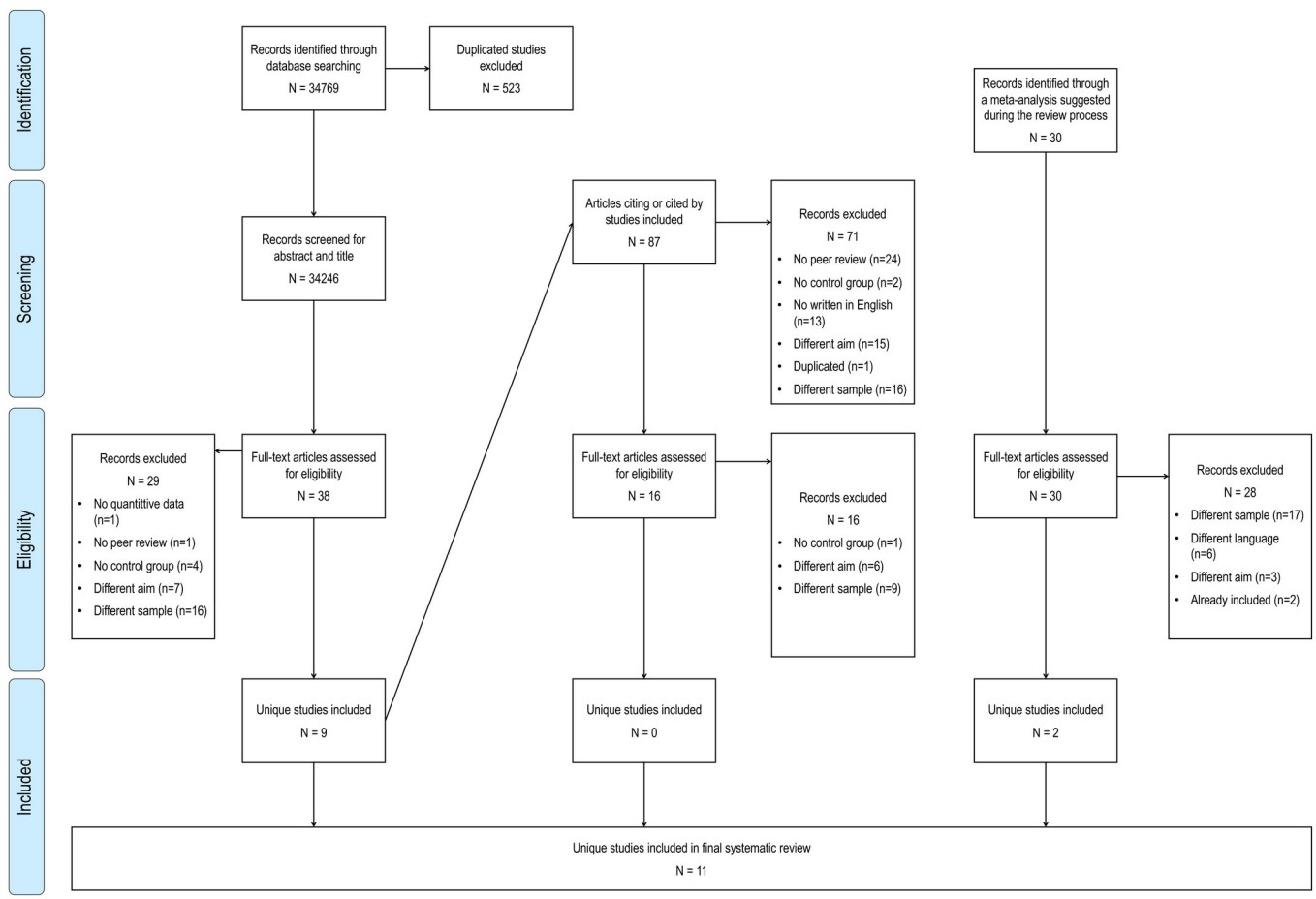

**Fig 1. PRISMA flowchart.** Flow of information through the different phases completed in the systematic review.

items of Part B are grouped into seven categories. Item b1 belongs to the category "Realism of the matter raised", Item b2 to "Existence or not of a final product", Item b3 to "Inclusion or not of group work", Items b4 to b7 to "Scaffolding by the teacher during learning", Item b8 to "Autonomy granted to students when making decisions about the project", Items b9 to b12 to "Correct evaluation tools employed", and Items b13 to b16 to "Explicit practice of metacognitive skills". The categories which conform the scale were elaborated based on the principles suggested by reference review works in this field [5,7,9,11,14]. For the sake of consistency, each category in the scale must have been mentioned by at least two of these reference sources. As in the quality scale mentioned above, each item could obtain one of three values. MF and SPL independently scanned all the studies and assigned a value to each item, reaching an initial agreement of 99.63%. Disagreements were resolved by discussion and consensus between the two researchers until there was 100% agreement. Fig 3 shows a detailed description of the values assigned to each item.

## Results

### Description of the studies

Table 1 provides a detailed summary of the eleven studies included in this review. Overall, many of the coded elements showed substantial heterogeneity, such as sample size, age of participants, duration of the interventions, or reported outcomes. Specifically, the total sample

Fig 2. Scale of quality and values assigned to each item. Summary of the items which comprise the scale of quality and values assigned to each of them in each study.

consisted of 722 participants, aged between 6 and 11 years. Among them, 101 were kindergarten students and 621 were first- to sixth-grade students. The interventions lasted between 4 and 32 weeks. Some of the subjects covered were science, mathematics, or English.

As can be seen in Table 1, most of the studies included in this review reported positive effects of PjBL on academic achievement. More precisely, six studies showed significant improvement of students trained through PjBL compared to students trained with other methods; three studies obtained improvements through both methods (without making comparisons between the experimental and control groups); one study found no significant difference between PjBL and other types of training; and the remaining study reported no quantitative data.

## Quality scale

Fig 2 shows the results of the qualitative assessment of the eleven studies included in the review. Across all items, 28.98% were rated as positive, 41.48% as negative, and 29.55% as unknown.

**Fig 3. Scale of prerequisites and intervention fidelity.** Summary of the prerequisites necessary for students and teachers for a successful PjBL adoption and intervention fidelity criteria in light of the key elements of the method.

| | | Alacapinar, 2008 | Aral et al., 2010 | Aslan, 2013 | Cakici et al., 2013 | Can et al., 2017 | Gültekin, 2005 | Hastie et al., 2017 | Karacalli et al., 2014 | Kuharski et al., 2005 | Lin, 2015 | Zumbach et al., 2004 |
|---|---|---|---|---|---|---|---|---|---|---|---|---|
| a. Prerequisites | a1. Be able to discuss ideas | ? | ? | ? | ? | ? | ? | ? | ? | ? | ? | + |
| | a2. Communicate clearly | ? | ? | ? | ? | ? | ? | ? | ? | ? | ? | ? |
| | a3. Consider alternatives systematically | ? | ? | ? | ? | ? | ? | ? | ? | ? | ? | ? |
| | a4. Monitor their own understanding | ? | ? | ? | ? | ? | ? | ? | ? | ? | ? | ? |
| | a5. Compare their point of view with that of others | ? | ? | ? | ? | ? | ? | ? | ? | ? | ? | ? |
| | a6. Ask clear questions. | ? | ? | ? | ? | ? | ? | ? | ? | ? | ? | ? |
| | a7. Measurement of prior knowledge and skills | ? | ? | + | ? | ? | ? | ? | + | ? | ? | + |
| | a8. Their role as facilitators | ? | ? | ? | ? | ? | ? | ? | ? | ? | ? | ? |
| | a9. How much scaffolding to provide | ? | ? | ? | ? | ? | ? | ? | ? | ? | ? | ? |
| | a10. How to support students | ? | ? | ? | ? | ? | ? | ? | ? | ? | ? | ? |
| | a11. How to challenge student's thinking | ? | ? | ? | ? | ? | ? | ? | ? | ? | ? | ? |
| | a12. How to ask open-ended questions | ? | ? | ? | ? | ? | ? | ? | ? | ? | ? | ? |
| | a13. Appropriate methods of assessment | ? | ? | ? | ? | ? | ? | ? | ? | ? | ? | ? |
| | a14. How to monitor progress of students | ? | ? | ? | ? | ? | ? | ? | ? | ? | ? | ? |
| b. Intervention fidelity | b1. An authentic, real-world question | ? | ? | + | ? | + | ? | + | + | + | + | + |
| | b2. A final product | ? | ? | + | + | + | + | + | + | ? | + | + |
| | b3. Group work | + | ? | ? | + | + | + | + | + | ? | + | + |
| | b4. Provision of suitable resources | ? | ? | + | + | ? | + | + | + | ? | + | ? |
| | b5. Provision of relevant reading or on-line materials | ? | ? | ? | ? | ? | ? | ? | + | ? | + | ? |
| | b6. Adequate guidance for students | ? | ? | ? | ? | ? | + | + | + | ? | + | + |
| | b7. Modelling students | ? | ? | ? | ? | ? | ? | ? | ? | ? | ? | ? |
| | b8. Student-driven | + | ? | ● | + | + | ? | + | + | + | ● | + |
| | b9. Journal or notebook entries | ? | ? | ? | ? | ? | ? | ? | ? | ? | + | ? |
| | b10. Portfolio assessment | + | ? | ? | ? | ? | ? | + | + | + | ? | ? |
| | b11. Clinical interviews | + | ? | ? | ? | + | + | ? | ? | ? | ? | ? |
| | b12. Examining student discourse | ? | ? | ? | ? | ? | ? | ? | ? | ? | ? | ? |
| | b13. Generate plans | + | ? | ? | ? | + | ? | + | + | ? | ? | ? |
| | b14. Systematically generate and test predictions | ? | ? | ? | ? | ? | ? | ? | + | ? | ? | ? |
| | b15. Interpret evidence in light of those predictions | ? | ? | ? | ? | ? | ? | ? | + | ? | ? | ? |
| | b16. Determine solutions | + | ? | ? | + | + | ? | + | + | ? | + | + |

Most of the studies followed a quasi-experimental design (Items 2 and 3) and did not include an active control group (Item 10). None of the studies guaranteed blinding of participants, instructors, and evaluation or, alternatively, did not report any information on this matter (Items 4 to 6). Similarly, most of the studies failed to provide enough information to replicate the intervention or the dependent variable (Items 11 and 12), and no study informed about the validity of the latter (Item 14). None of the studies had been preregistered or made the data publicly available on the Internet (Items 1 and 16). In contrast, most of the studies confirmed the similarity of the experimental and control groups in terms of socio-economic characteristics (Item 7). Likewise, most of the studies reported the analysis of pre-test scores in experimental and control groups (Item 8) and analyzed the differences between them (Item 15).

## Prerequisites and intervention fidelity scale

Fig 3 shows a summary of the information related to the prerequisites and intervention fidelity of the studies. As can be seen, overall 20.61% of the items obtained positive values, 0.61% obtained negative values, and 78.79% were labeled as unknown. Most studies offered little or no information to assess the items related to compliance with prerequisites (Part A). Only one study reported specific information about the training of students in group work (Items a1 to a6) and it focused exclusively on the ability to discuss ideas (Item a1). Similarly, just two studies informed about the training of teachers in PjBL (Item a8 to a14) but none of them provided any information about the content of this training and, consequently, they were coded as "unknown". Finally, information related to prior knowledge of students before starting the project was reported in three studies (Item a7).

In comparison, the studies reported more information about the intervention fidelity (Part B). Overall, 51.82% of the items obtained positive values, 1.82% obtained negative values, and 46.36% were labeled as unknown. Items b1-b3, coding for the realism of the problem, the existence of a final product, and the inclusion of group work were relatively well reported and received positive scores. Within the items focused on scaffolding, Items b4 and b6 were met by more than half of the studies, while Item b5 was only reported by two studies and Item b7 was not addressed in any study. Item b8, related to the autonomy provided to students, was well reported by more than half of the studies, but, importantly, two of them received negative scores. The rest of the items related to the appropriateness of evaluation tools (Items b9-b12) and to the explicit practice of meta-cognitive skills (Items b13-b16) were generally reported with insufficient detail, except for Item b16, where 6 studies obtained positive scores. Overall, the information about intervention fidelity was often reported too vaguely and had to be inferred indirectly from information scattered throughout the papers. For example, in the study of Alacapinar (2008) [37], Item b13, related to planning skills, was inferred on the basis of the following statement: "[Students] learned by experience how important it is to plan work and accomplish it in a given time" (p. 28).

## Discussion

PjBL is a student-centered methodology that promotes the acquisition of higher-order thinking skills thought the solution of real problems in collaborative groups and with limited guidance of the teacher [6,9,10]. Although this approach has become the cornerstone of innovative movements in many schools [49,50], the evidence supporting its effectiveness in the classroom is still scarce [5,51]. The objective of the present review was to assess the available evidence about the impact of PjBL on the acquisition of content knowledge by kindergarten and primary students.

The articles aimed at examining the impact of PjBL in kindergarten and primary students were scarce and, overall, yielded mixed results. Specifically, seven of the 11 studies included in

this review obtained positive results regarding the impact of PjBL in academic achievement of students immediately after the intervention [36,38,41,42,46,47] or in the long term [44]. Among the rest, two studies did not find significant differences between the experimental and control groups [37,43], one study did not report any quantitative data [40], and another one offered no comparison between the experimental and control groups [39]. In addition, the studies showed considerable heterogeneity in terms of participants' age (from preschoolers to 11 years old students), duration of the intervention (ranging from 3 to 32 weeks), and measured outcomes (e.g., categorization skills or English knowledge). This hinders the generalization of the results to the entire school population.

Along with the mixed results obtained, an in-depth analysis of the studies showed important shortcomings that deserve more attention in future research. Firstly, there is room for improvement in the methodological quality of the studies. For instance, none of them followed an experimental design and only two included an active control group. These deficiencies make it hard to draw meaningful conclusions from the results. In fact, if all of these results had been collated in a quantitative meta-analysis without a proper analysis of their quality, most likely the conclusions would have been deceivingly positive. Without more and better evidence, it is difficult to assess whether PjBL is effective for kindergarten and elementary school students.

Secondly, information concerning important aspects of the materials and procedure was usually not reported or, when reported, revealed suboptimal methods, compromising the replicability of the studies. For example, many authors did not provide information about the tests used to measure the outcomes, the specific activities performed, or the intervention materials. Even if PjBL was successful, in all these cases it would be impossible to bring the intervention proposals to the classroom. This becomes more concerning if we consider the lack of a universally accepted model of PjBL [14]. In the same vein, none of the studies granted access to the data, which means that the reproducibility of the results cannot be verified by independent researchers.

Thirdly, few studies reported sufficient information to ensure that the interventions met the necessary requirements from students and instructors to guarantee the success of PjBL. For instance, only three studies measured prior knowledge of students before the intervention, only one offered information about students' training in group work, and none described the training of instructors in PjBL, had it existed. The importance of these elements is often highlighted in the literature [5,7,9,15,52]. Precisely, a recent literature review by [33] highlighted two of them as essential for the success of PjBL: effective group work of pupils and support to teachers through regular networking and professional development opportunities. Given the lack of detailed information on these aspects in the studies included in this review, it is impossible to weight the contribution of these factors to the final results.

Finally, regarding intervention fidelity, although several of the key components of PjBL were well covered in the majority of studies (e.g., the use of real word-problems, the elaboration of a final product, or collaborative work), others were broadly neglected (e.g., the amount of guidance provided to students during the intervention, the evaluation tools used by teachers, or the training on metacognitive skills). As in the case of the prerequisites mentioned above, a considerable volume of research has stressed the importance of considering these elements in PjBL, including the monitoring of students [10,15,33,53] or the employment of adequate assessment tools to measure the progress of pupils [11,33]. These information gaps impede to determine what is decisive in this kind of intervention to be effective and, at the same time, hamper the distinction between PjBL and other educational interventions. This concern has been raised in previous reviews [21,54].

## Classroom implications and future research

PjBL provides highly desirable benefits for students, such as the creation of independent, self-regulated learners [12,55,56], the promotion of engagement towards learning [9,50,57–60], or the fostering of meaningful learning [50,61]. However, more and better evidence is needed about how and when PjBL is most suitable. For the purpose of this review, it is relevant to consider that the majority of studies assessing the impact of PjBL on learning have been aimed at higher education students [9]. But what is effective in a secondary or a postsecondary setting may not transfer directly to kindergarten and primary students [25]. It would be convenient to reflect on the suitability of this approach for younger students. Specifically, it should not be assumed that novice learners possess the advanced self-regulation skills, prior knowledge, or group work skills (for example, the abilities needed to discuss ideas, consider alternatives, or compare different points of view) necessary for PjBL [7,9,36]. Hence, any attempt to translate the main results and conclusions of this literature to kindergarten and primary students should be properly monitored. Apart from the educational stage, little is known about how different learning profiles might make PjBL more or less effective, as in the case of learners with different educational background or those with learning disabilities [9,14,25]. The studies included in this review do not contribute to this question, since the population of all them is composed by students without special needs.

Last but not least, we think that future research in this domain should try to overcome the shortcomings we encountered in conducting this review. This includes the lack of active control groups; the lack of randomly assigned participants; inappropriate blinding of participants, instructors, and evaluators; un-validated measures for the learning outcomes; or the lack of detailed information to replicate the study (e.g., activities conducted, approximate duration of each session, or evaluation tools used). Future research should also address the impact of some basic prerequisites by students (e.g., group work) and teachers (e.g., evaluation tools) on PjBL intervention. Similarly, it would be advisable that researches provide detailed information about the fidelity of the intervention to key features of PjBL [11,14,33]. These important shortcomings should also be taken into consideration in interpreting or applying already published interventions.

The academic success of many students, especially those with learning difficulties, depends largely on the use of methods that have proven to be consistently effective [62]. For this to be possible, the incorporation of research findings into decision making process, along with the tacit knowledge, values, and thoughts of educators, becomes indispensable. The adoption of this approach, known as research-informed practice, is a daunting challenge and involves many different actors and stakeholders [63,64]. In view of the above, researchers can surely contribute to this aim by providing more and better evidence on the conditions under which PjBL is effective.

## Limitations of the present review

The main limitation of the present review is the scarce number of studies found. From the almost 40 full-text articles initially screened for eligibility, just nine met the selection criteria. Similarly, from the 30 studies contained in the meta-analysis suggested by one anonymous reviewer, just two were finally added. In light of this, it is difficult to draw firm conclusions about the effectiveness of PjBL for kindergarten and elementary school students, beyond highlighting shortcomings that should be addressed in future research. A wider search for studies, perhaps not limited to peer-reviewed articles (such as papers presented at conferences), might have yielded more results, although this option would most likely diminish the average quality of the final sample of studies. Besides, given that PjBL is particularly

recommended for the development of domain-general skills [5,9], it would have been interesting to test the impact of PjBL not only on academic achievement but also on the development of higher-order skills, such as problem solving, critical thinking, deep understanding, or self-evaluation of students.

## Supporting information

**S1 checklist.**
(DOC)

## Author Contributions

**Conceptualization:** Marta Ferrero, Samuel P. León.

**Formal analysis:** Marta Ferrero, Samuel P. León.

**Methodology:** Marta Ferrero, Miguel A. Vadillo, Samuel P. León.

**Writing – original draft:** Marta Ferrero, Samuel P. León.

**Writing – review & editing:** Marta Ferrero, Miguel A. Vadillo, Samuel P. León.

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
