## [Editor Report · Decision Letter 0]

22 Jul 2020

PONE-D-20-21356

Is project-based learning effective among kindergarten and elementary students? A systematic review

PLOS ONE

Dear Dr. Vadillo,

Thank you for submitting your manuscript to PLOS ONE. After careful consideration, we have decided that your manuscript does not meet our criteria for publication and must therefore be rejected.

Specifically:

Although the criteria have mostly been met, for example, methods and analyses are performed to a high technical standard and are described in sufficient detail (PRISMA), Criterion 4 (discussion is presented in an appropriate fashion and is supported by the data) needs major work. My suggestions for improvement before sending the proposal to another journal are: 

1) Both introduction and discussion sections should be supported by more recent papers, ideally from the WoS system. There were few articles meeting these high standards (e.g., Hasni et al., 2016). 

2) Literature predicting small effectiveness of unguided PBL should be cited to make a more compelling case supporting the results, such as:

- Mayer, R. E. (2004). Should there be a three-strikes rule against pure discovery learning? American Psychologist, 59(1), 14-19. doi: 10.1037/0003-066x.59.1.14

- Kirschner, P. A., Sweller, J., & Clark, R. E. (2006). Why minimal guidance during instruction does not work: An analysis of the failure of constructivist, discovery, problem-based, experiential, and inquiry-based teaching. Educational Psychologist, 41(2), 75-86. doi: 10.1207/s15326985ep4102_1

I am sorry that we cannot be more positive on this occasion, but hope that you appreciate the reasons for this decision.

Yours sincerely,

Juan Cristobal Castro-Alonso, Ph.D.

Academic Editor

PLOS ONE

- - - - -

---

## [Author Response · Author response to Decision Letter 0]

4 Dec 2020

Comment from the editor: Literature predicting small effectiveness of unguided PBL should be cited to make a more compelling case supporting the results, such as:

- Mayer, R. E. (2004). Should there be a three-strikes rule against pure discovery learning? American Psychologist, 59(1), 14-19. doi: 10.1037/0003-066x.59.1.14

- Kirschner, P. A., Sweller, J., & Clark, R. E. (2006). Why minimal guidance during instruction does not work: An analysis of the failure of constructivist, discovery, problem-based, experiential, and inquiry-based teaching. Educational Psychologist, 41(2), 75-86. doi: 10.1207/s15326985ep4102_1

Response: Thank you for your suggestions. We have added both articles in the Discussion.

Comment from the editor: Both introduction and discussion sections should be supported by more recent papers, ideally from the WoS system. There were few articles meeting these high standards (e.g., Hasni et al., 2016). 

Response: The systematic review run in WoS, PsycInfo, and ERIC does not yield more articles than the ones included in the previous version.

---

## [Decision Letter · Decision Letter 1]

25 Jan 2021

PONE-D-20-21356R1

Is project-based learning effective among kindergarten and elementary students? A systematic review

PLOS ONE

Dear Dr. Vadillo,

Thank you for submitting your manuscript to PLOS ONE. After careful consideration, we feel that it has merit but does not fully meet PLOS ONE’s publication criteria as it currently stands. Therefore, we invite you to submit a revised version of the manuscript that addresses the points raised during the review process.

We look forward to receiving your revised manuscript.

Kind regards,

Mingming Zhou, Ph.D.

Academic Editor

PLOS ONE

Journal Requirements:

(2) Please include captions for your Supporting Information files at the end of your manuscript, and update any in-text citations to match accordingly. Please see our Supporting Information guidelines for more information: http://journals.plos.org/plosone/s/supporting-information.

Reviewers' comments:

Reviewer's Responses to Questions

**Comments to the Author**

1. If the authors have adequately addressed your comments raised in a previous round of review and you feel that this manuscript is now acceptable for publication, you may indicate that here to bypass the “Comments to the Author” section, enter your conflict of interest statement in the “Confidential to Editor” section, and submit your "Accept" recommendation.

Reviewer #1: All comments have been addressed

Reviewer #2: All comments have been addressed

2. Is the manuscript technically sound, and do the data support the conclusions?

Reviewer #1: Yes

Reviewer #2: Partly

3. Has the statistical analysis been performed appropriately and rigorously? 

Reviewer #1: Yes

Reviewer #2: N/A

4. Have the authors made all data underlying the findings in their manuscript fully available?

Reviewer #1: Yes

Reviewer #2: Yes

5. Is the manuscript presented in an intelligible fashion and written in standard English?

Reviewer #1: Yes

Reviewer #2: Yes

6. Review Comments to the Author

Reviewer #1: 1.Please write the year of the selection paper in both the abstract and the manuscript.

2.Please write the research sample background (outlining in which areas).

3.Please describe the originality/value of this research.

4.Please detail the effectiveness of project-based learning for kindergarten and elementary school students regarding their academic performance in the introduction.

Reviewer #2: This paper reports on an interesting topic, and I believe this research can contribute to a better understanding of project-based learning in K–6 practices. Although this paper, in general, is well written, several problems or issues in this paper need to be addressed.

1. The authors could consider using “PjBL” to refer to project-based learning throughout the manuscript, because “PBL” often refers to problem-based learning in the literature.

2. I was wondering whether PjBL is suitable for kindergarten students.

3. For the topics of effects of PjBL, I would suggest the authors to review a meta-analysis article: https://doi.org/10.1016/j.edurev.2018.11.001 , for greater understanding of its effects by cutting-edge research.

4. This research only includes 9 articles; however, it seems that the above meta-analysis included 8 elementary school PjBL studies (many are different from yours). I suggest checking those studies (to include more articles).

5. I suggest searching more databases such as EBSCOhost and ProQuest.

6. There is a gap between the 34,246 studies and the 38 studies (p. 7).

7. High resolution figures are needed; kindly provide high-resolution original figures (Figures 1–3) in the manuscript.

7. PLOS authors have the option to publish the peer review history of their article (what does this mean?). If published, this will include your full peer review and any attached files.

Reviewer #1: No

Reviewer #2: No

---

## [Author Response · Author response to Decision Letter 1]

13 Feb 2021

Please, find an attached document with our response to the reviewers.

---

## [Decision Letter · Decision Letter 2]

23 Mar 2021

Is project-based learning effective among kindergarten and elementary students? A systematic review

PONE-D-20-21356R2

Dear Dr. Vadillo,

We’re pleased to inform you that your manuscript has been judged scientifically suitable for publication and will be formally accepted for publication once it meets all outstanding technical requirements.

Kind regards,

Mingming Zhou, Ph.D.

Academic Editor

PLOS ONE

Additional Editor Comments (optional):

Reviewers' comments:

Reviewer's Responses to Questions

**Comments to the Author**

1. If the authors have adequately addressed your comments raised in a previous round of review and you feel that this manuscript is now acceptable for publication, you may indicate that here to bypass the “Comments to the Author” section, enter your conflict of interest statement in the “Confidential to Editor” section, and submit your "Accept" recommendation.

Reviewer #1: All comments have been addressed

Reviewer #2: All comments have been addressed

2. Is the manuscript technically sound, and do the data support the conclusions?

Reviewer #1: Yes

Reviewer #2: Yes

3. Has the statistical analysis been performed appropriately and rigorously? 

Reviewer #1: Yes

Reviewer #2: Yes

4. Have the authors made all data underlying the findings in their manuscript fully available?

Reviewer #1: Yes

Reviewer #2: Yes

5. Is the manuscript presented in an intelligible fashion and written in standard English?

Reviewer #1: Yes

Reviewer #2: Yes

6. Review Comments to the Author

Reviewer #1: The authors improved the article in response to the reviews. I agree to the publication of the paper submitted to the journal.

Reviewer #2: The authors have made appropriate responses and/or revisions to the comments made by the reviewers. I think the revised version of this article is ready for publication.

7. PLOS authors have the option to publish the peer review history of their article (what does this mean?). If published, this will include your full peer review and any attached files.

Reviewer #1: No

Reviewer #2: No

---

## [Editor Report · Acceptance letter]

25 Mar 2021

PONE-D-20-21356R2 

Is project-based learning effective among kindergarten and elementary students? A systematic review 

Dear Dr. Vadillo:

I'm pleased to inform you that your manuscript has been deemed suitable for publication in PLOS ONE. Congratulations! Your manuscript is now with our production department. 

Kind regards, 

on behalf of

Dr. Mingming Zhou 

Academic Editor

PLOS ONE